# Enhancing Oral Delivery of Biologics: A Non-Competitive and Cross-Reactive Anti-Leptin Receptor Nanofitin Demonstrates a Gut-Crossing Capacity in an Ex Vivo Porcine Intestinal Model

**DOI:** 10.3390/pharmaceutics16010116

**Published:** 2024-01-16

**Authors:** Solene Masloh, Anne Chevrel, Maxime Culot, Anaëlle Perrocheau, Yogeshvar N. Kalia, Samuel Frehel, Rémi Gaussin, Fabien Gosselet, Simon Huet, Magali Zeisser Labouebe, Leonardo Scapozza

**Affiliations:** 1Blood Brain Barrier Laboratory, Faculty of Science Jean Perrin, Artois University, UR 2465, Rue Jean Souvraz, 62300 Lens, Francemaxime.culot@univ-artois.fr (M.C.); fabien.gosselet@univ-artois.fr (F.G.); 2Affilogic, 24 Rue de la Rainière, 44300 Nantes, Franceanaelle@affilogic.com (A.P.); remi@affilogic.com (R.G.); 3School of Pharmaceutical Sciences, University of Geneva, 1 Rue Michel Servet, 1201 Geneva, Switzerlandleonardo.scapozza@unige.ch (L.S.); 4Institute of Pharmaceutical Sciences of Western Switzerland, University of Geneva, 1 Rue Michel Servet, 1201 Geneva, Switzerland

**Keywords:** biotherapeutics, permeability, shuttle, leptin receptor, Nanofitins, ex vivo, porcine intestinal model, receptor-mediated transport, oral-to-systemic delivery

## Abstract

Biotherapeutics exhibit high efficacy in targeted therapy, but their oral delivery is impeded by the harsh conditions of the gastrointestinal (GI) tract and limited intestinal absorption. This article presents a strategy to overcome the challenges of poor intestinal permeability by using a protein shuttle that specifically binds to an intestinal target, the leptin receptor (LepR), and exploiting its capacity to perform a receptor-mediated transport. Our proof-of-concept study focuses on the characterization and transport of robust affinity proteins, known as Nanofitins, across an ex vivo porcine intestinal model. We describe the potential to deliver biologically active molecules across the mucosa by fusing them with the Nanofitin 1-F08 targeting the LepR. This particular Nanofitin was selected for its absence of competition with leptin, its cross-reactivity with LepR from human, mouse, and pig hosts, and its shuttle capability associated with its ability to induce a receptor-mediated transport. This study paves the way for future in vivo demonstration of a safe and efficient oral-to-systemic delivery of targeted therapies.

## 1. Introduction

Recombinant biological molecules, the powerful pioneers of modern medicine, referred to as biologics—encompassing entities such as antibodies, proteins, and peptides—constitute the very backbone of contemporary medical treatments [1,2,3]. However, unlike conventional small-molecule drugs commonly administered orally [4], they are mostly limited to parenteral injections mainly due to their sensitivity to protease degradation and their limited passive diffusion through the intestinal barrier [1]. Specifically, the poor permeability of biologics across the intestinal barrier is attributed to the presence of tight junctions at the border of intestinal epithelial cells which hinder their paracellular transport [5], as well as to the enzymes and transporters expressed by these cells which limit their transcellular passage to specific active mechanisms [6,7,8,9]. While the stability in the GI tract of biologics can be solved by gastro-resistant formulation, enabling the permeability of biologics through the intestinal barrier remains a critical factor in achieving successful oral-to-systemic delivery of targeted biologics-based therapies.

Harnessing the active translocation mechanisms employed by receptors for biomolecule passage through biological barriers has shown promise for drug transport in preclinical studies, particularly across the blood–brain barrier (BBB) that isolates the brain from the bloodstream [10,11,12]. This strategy, known as molecular trojan horse or receptor-mediated transcytosis (RMT), involves targeting various receptors of natural ligands, including the transferrin receptor [13,14], the low-density lipoprotein receptor [14,15], and the insulin receptor [12,16], which can trigger the internalization and transport of their natural ligand across biological barriers. Adopting a similar strategy to facilitate the translocation of biologics across the intestinal barrier could enable their systemic delivery after oral administration.

Several receptors, including the neonatal Fc receptor [17,18] and LepR [19,20], have been partially characterized for their involvement in the active transport of protein-based ligands across the intestinal epithelium, despite the scarcity of information available in this field. The LepR, a member of the class I cytokine receptor family, exists in various isoforms resulting from alternative splicing [21,22] (long, short, or secretory). The membrane-bound isoforms share common extracellular and transmembrane domains, whereas their intracellular regions differ in size and amino-acid sequence. They have an identical affinity for the leptin since the ligand-binding region is located on the shared extracellular segment [23]. Evidence suggests that LepR facilitates a specific, saturable, and energy-dependent process for the transcytosis of leptin across the intestinal barrier [19,24]. Leptin interacts with the extracellular domain of the receptor present on the apical membrane of enterocytes and is internalized by the cells through their endosomal compartments. The leptin-LepR complex is then packaged and discharged on the basolateral membrane to reach the blood circulation across the capillary endothelial cells [24]. Given its biological significance, LepR presents itself as a compelling receptor for RMT.

Our present strategy to enhance the transport of biologics across the intestinal barrier relies on Nanofitins targeting the LepR (anti-LepR Nanofitins), which offer a combination of protein robustness, modularity, and tunable specificity. These small single-chain binding proteins of 7 kDa, devoid of cysteine residues, are derived from the naturally hyper thermostable protein Sac7d [25,26], or more generally from the Sul7d family [27]. They can be custom-engineered for high specificity and affinity toward a target, as already demonstrated against a variety of biological targets [28,29,30,31]. Importantly, they maintain the stability of their parental protein, including resistance to extreme pH levels (pH 1–13) and the harsh intestinal environment [28]. With their N- and C-termini located on opposite faces of their binding site, they can be easily assembled to a cargo molecule, either via chemical conjugation or genetic fusion, while preserving their individual properties [32].

In this proof-of-concept study, we investigate the potential of employing a delivery strategy based on anti-LepR Nanofitins as molecular shuttles to enhance the transport of biologics through the intestine. In vitro, our approach involved developing Nanofitins targeting the LepR. The target product profile included non-competition with the leptin binding to its receptor, aimed at improving safety by minimizing competition with endogenous ligands, and cross-reactivity with receptors from human, mouse, and pig hosts to suit relevant ex vivo and in vivo preclinical models. Ex vivo, we successfully demonstrated the potential of an anti-LepR Nanofitin candidate as an efficient carrier for transporting functional cargo proteins across the intestinal barrier, suggesting the potential of the NF scaffold for application in oral biologics delivery strategies.

## 2. Materials and Methods

### 2.1. Reagents

Horseradish peroxidase(HRP)-conjugated RGS-His antibody used for detection of Nanofitins was part of a detection kit from Qiagen (Hilden, Germany, cat. no. 34460). Rabbit monoclonal antibody against the LepR (cat. no. MA5-35247) and goat anti-rabbit secondary antibody (cat. no. 35565) used for Western blot were purchased from Thermo Fischer Scientific (Waltham, MA, USA). Pierce™ protein-free blocking buffer (cat. no. 37585), 3,3′,5,5′-tétraméthylbenzidine (TMB) for ELISA revelation (cat. no. 34022), Pierce Bicinchoninic acid (BCA) protein assay kit used to determine concentrations of lysates or fluorescently labeled Nanofitins (cat. no. 23225), DMEM with GlutaMAX™ Supplement (cat. no. 10566016), fetal calf serum (cat. no. 16010167), penicillin–streptomycin (cat. no. 15140122), and finally, the lipofectamine™ 2000 used for transfections (cat. no. 11668019) were also from Thermo Fischer Scientific. Recombinant human leptin (cat. no. 398-LP), recombinant human LepR (rhLepR), and recombinant mouse LepR (rmLepR) (cat. no. 389-LR-100/CF and cat.no. 497-LR-100/CF), the last two being expressed as a Fc fusion, were from R&D systems (Minneapolis, MN, USA), as well as the Alexa-Fluor 647-conjugated anti-LepR antibody (cat. no. FAB867P) and the Alexa-fluor 647-conjugated anti-His-tag antibody (cat. no. IC0501R) used for cellular assays. N-(5-Fluoresceinyl)-maleimide, human serum albumin (HSA, cat. no. SRP6182), phosphatase (cat. no P2850 and cat. no P5726)/protease (cat. no P8340) inhibitor cocktails, and Sera-mag SpeedBeads Protein A/G (cat. no. GE17152104010150) were obtained from Sigma-Aldrich (St.-Louis, MO, USA). HepG2 (cat. no. ab166833), HEK293 (cat. no. ab7902)m and HeLa (cat. no. ab150035) cell lysates were from Abcam (Cambridge, United Kingdom). Finally, the accutase solution (cat. no. SCR005) for cellular assays and the BugBuster^®^ Protein Extraction Reagent (cat. no. 70584) were obtained from Merck (Kenilworth, NJ, USA).

### 2.2. In Vitro Selection by Ribosome Display and Preliminary Screening

#### 2.2.1. In Vitro Selection by Ribosome Display

A combinatorial library of Nanofitins was generated following the protocol described in the literature with minor modifications [25,33]. In brief, the library was prepared with two successive overlapping PCRs using degenerated oligonucleotides at randomized positions encoded by trimer codons of naturally occurring amino acids (except cysteine). The 5′- and 3′-flanking regions necessary for the in vitro technique were then supplemented with a final PCR [34]. After the amplification of the library, a transcription was realized to start the in vitro selection process.

The ribosome display procedure was performed at 4 °C, following the protocol described by Mouratou et al. [25,33], with the rhLepR chimera (recombinant extracellular domain of the hLepR fused to the Fc portion of human IgG1) as the target of interest. The target was not biotinylated as described in the initial protocol since we took advantage of the Fc fusion to perform an immobilization on protein A beads. Four rounds of selection have been performed to isolate high affinity binders. The intensive washing was gradually increased to apply a higher selection pressure (round 1: eight washes of 20 s; round 2: eight washes of 3 min; round 3: eight washes of 15 min; and round 4: four washes of 15 min followed by four washes of 30 min).

#### 2.2.2. Isolation of Clones

The amplified DNA sequences recovered from the last round of selection were cloned between BamHI and HindIII restriction sites in a modified pQE-30 expression vector (Qiagen, Hilden, Germany) called pAFG12. The cloning of the DNA material inside this plasmid enabled the expression of Nanofitins with *Aequorea coerulescens* green fluorescent protein (acGFP) fused at their C-terminus using the linker sequence GSAGSAAGSGEF. *Escherichia coli* DH5α LacIq strain (Invitrogen, Waltham, MA, USA) was then transformed with the ligation mixtures using the manufacturer’s protocol. Following bacteria transformation, 95 fluorescent clones selected on yeast extract tryptone (2YT)-agar plates with ampicillin (100 µg/mL) and kanamycin (25 µg/mL) were picked and seeded into a 96-deep-well plate filled with 0.75 mL of 2YT medium complemented with antibiotics and 1% glucose in each well. After an overnight culture at 37 °C under shaking (600 rpm), 200 µL of each culture were used to inoculate in mirror another 96-deep-well plate containing 1.25 mL of 2YT medium supplemented with ampicillin (100 µg/mL), kanamycin (25 µg/mL), and 0.1% glucose. Also, 10 µL of the same precultures were deposed on a 96-well agar plate with ampicillin at 100 µg/mL for sequencing using the oligo Qe30_for CTTTCGTCTTCACCTCGA (Plateseq Service, Eurofins). The deep-well plate was incubated during 3 h at 37 °C and 600 rpm before the addition of 50 μL of IPTG to reach a concentration of 0.5 mM (isopropyl β-D-1-thiogalactopyranoside, Sigma-Aldrich, cat. no. I5502) used to induce the expression of Nanofitins. This step was followed by another incubation under shaking (600 rpm) at 30 °C for 4 h. After induction, cells were pelleted via a centrifugation of 20 min at 2000× *g* and the supernatants were discarded. Then, Nanofitins were extracted via cell lysis with 100 µL/well of BugBuster protein extraction reagent (Novagen, Madison, WI, USA, cat. no. 70584) for 1 h at room temperature (RT). Finally, 350 μL of tris-buffered saline (TBS, 20 mM Tris-HCl, 150 mM NaCl, pH 7.4) were added and cell debris were pelleted by 20 min of centrifugation at 2000× *g*. The supernatants were used to perform a first screening of Nanofitins generated against the rhLepR.

#### 2.2.3. Preliminary Screening by ELISA

Enzyme-linked immunosorbent assays (ELISA) were performed according to standard protocols. Briefly, rhLepR diluted at 5 μg/mL in TBS (100 µL/well) was immobilized on 96-well flat-bottomed Nunc MaxiSorp plates (Thermo Fischer Scientific) incubated at 4 °C overnight or 1 h at RT. Plates were then washed 3 times with TBS (300 µL/well) and blocked for 1 h at RT under shaking (600 rpm) with 300 µL/well of TBS-bovine serum albumin 0.5% (*w*/*v*) (BSA, Sigma-Aldrich). Each of the following steps (except revelation) was followed by an incubation of 1 h at RT and 4 washes with TBS-Tween 20 0.1% (TBST). After the blocking, 100 µL of crude *E. coli* extracts were added to the wells with or without the immobilized target. In order to detect specific binding of Nanofitins, an HRP-conjugated RGS-His antibody diluted 4000 times in TBST was used. The revelation was performed 1 h later with TMB (3,3′,5,5′-tetramethylbenzidine) and stopped after 5 min with the addition of H_2_SO_4_ solution (2 N, 50 µL/well). Finally, absorbance was measured at 450 nm on a Varioskan ELISA plate reader (Thermo Fischer Scientific).

### 2.3. Production of Nanofitins

Following four rounds of ribosome display, the most relevant anti-LepR Nanofitins were selected for subsequent investigations. This selection was based on an analysis of the sequences of the isolated clones and a preliminary screening of their affinity for the rhLepR. Plasmids coding for the most interesting candidates were ordered from Genscript (Piscataway, NJ, USA), performing gene synthesis and subcloning into a pet21a(+) vector between the NdeI/BlpI cloning site. The same approach for the Nanofitin fusions was undertaken with Genscript: gene synthesis of Nanofitin (NF1) and Nanofitin (NF2) linked by a nucleotide sequence coding for a (G_4_S)_2_ linker between NF1 and NF2 and subsequent subcloning. All protein constructs carry an RGS-His_6_ tag at the C-termini of the construct. Also, the most interesting candidates chosen to perform the ex vivo study following the characterization, as well as the Nanofitin fusions, carry a unique cysteine at the N-termini of the construct to be able to perform a regioselective conjugation to a fluorescent marker. Nanofitins were expressed in BL21 Gold (DE3) strain of *E. coli* (Agilent technologies, Santa Clara, CA, USA) following a bacterial transformation. Their precultures, grown at 37 °C overnight in 2YT medium with ampicillin (100 μg/mL), tetracyclin (5 μg/mL), and 1% of glucose (*w*/*v*), were diluted 20 times in a similar medium with glucose at 0.1% (*w*/*v*). Cultures were then incubated at 37 °C until the mid-log phase. Once optical density at 600 nm reached 0.8 to 1, Nanofitins expression was triggered by addition of IPTG at 0.5 mM for 4 h at 30 °C. After 45 min of centrifugation at 3220× *g*, pelleted cells were resuspended in a lysis buffer (1× BugBuster Protein Extraction Reagent, 5 μg/mL DNaseI, 20 mM Tris, 500 mM NaCl, 25 mM imidazole, pH 7.4) at RT during 15 min under stirring. Following this lysis, another centrifugation of 45 min was carried out to eliminate cell debris. The His-tagged Nanofitins from the supernatants were then filtered (0.45 µm) and purified on an immobilized metal ion affinity chromatography using HisTrap HP column (Cytiva, Marlborough, MA, USA) or bulk Ni-NTA resin (Ozyme, 635660). Finally, Nanofitins were eluted with a buffer composed of 20 mM Tris, 500 mM NaCl, and 500 mM of imidazole, pH 7.4. Concentrations of the Nanofitins were determined from absorbance at 280 nm with a Biospectrometer (Eppendorf, Hamburg, Germany).

### 2.4. Biochemical Characterization of Nanofitins

#### 2.4.1. Binding Assays: Determination of the Affinity, Kinetics of Interaction, and the Competition Profile with Leptin

##### ELISA

ELISAs were performed according to the protocol described in Section 2.3. Either rhLepR (5 μg/mL), rmLepR (5 μg/mL), or HSA (2.5 μg/mL), diluted in TBS, were immobilized on 96-well flat-bottom Nunc MaxiSorp plates (Thermo Fischer Scientific). The blocking step was performed with Pierce™ Protein-Free (TBS) Blocking Buffer when the plates were coated with HSA. EC_50_ were determined by adding 100 µL/well of purified Nanofitins from 10^−12^ to 10^−5^ M after the blocking step, with 10-fold dilutions, with or without the immobilized target. Binding curves were fitted following the measurement of the absorbance at 450 nm on a Varioskan plate reader, and EC_50_ were calculated using GraphPad Prism 7.0 software (GraphPad Software, Inc., San Diego, CA, USA).

Competition experiments with the natural ligand were performed with purified Nanofitins at their maximum concentration, providing a linear ELISA signal (from 2 nM for the best affinities, to 5 µM for weaker binders) with leptin at 100 nM.

This methodology has also been used to evaluate the binding activity of Nanofitins after ex vivo transport experiments. Samples recovered from these assays were applied instead of the purified Nanofitins. The Nanofitin solutions of the initial donors (solutions in the donor chambers at the beginning of the experiments) were diluted at the same concentrations than the Nanofitins solutions recovered from the serosal chambers (receivers) after 120 min: 1000 and 660 times, respectively, for 1-F08 and 1-F08-NF2. For the dimeric construct, samples were tested separately against immobilized rhLepR and HSA.

##### Binding Kinetics Assays by Bio-Layer Interferometry

Additional characterization of Nanofitins interactions with the rhLepR was performed by bio-layer interferometry (BLI) on Octet RED96 system (ForteBio, Fremont, CA, USA). Prior to the experiments, protein A biosensors (ForteBio, cat. no. 18-5010) were rehydrated for 10 min in a 96-well plate filled with 200 µL of TBS and equilibrated in TBS-BSA 0.01%-Tween 20 0.002%. Unless specified, all steps were performed in this latter buffer. Biosensors were then functionalized with rhLepR diluted at 5 µg/mL until the threshold of 1 nm was reached. After an equilibration of 60 s in the buffer, associations were measured by exposing the sensors to a concentration range of Nanofitins (333, 111, 37, 12.3, 4.1, and 1.4 nM) alongside a condition without Nanofitin used as reference. For each protein, the sequence of association and dissociation steps (180 s each) was followed by a regeneration of the sensors with 3 cycles alternating 10 s in glycine (10 mM, pH 2) and 10 s in TBS. All steps were performed under continuous 1000 rpm shaking at 30 °C. Sensorgrams were analyzed with the Octet Data Analysis software 11.1 (ForteBio) and fitted with a 1:1 binding model providing K_D_, the equilibrium binding constant in M; k_on_, the association rate constant in M^−1^s^−1^; and k_off_, the dissociation rate constant in s^−1^.

#### 2.4.2. Cross-Reactivity: Cell Transfection and Flow Cytometry

HEK293 (human embryonic kidney 293, CRL-1573) cells, obtained from ATCC (Rockville, MD, USA), were grown at 37 °C in a humidified atmosphere (95% O_2_ and 5% CO_2_) and in DMEM with GlutaMAX™ supplemented with 10% of fetal calf serum and 1% of penicillin/streptomycin. Once cells reached 70–90% confluence, they were washed with PBS and collected with an accutase solution. HEK293 cells were then seeded at 2.1 × 10^6^ cells into 10 cm petri dishes. After 24 h, cells were transfected using the manufacturer’s protocol (Thermo Fischer Scientific) with 15 µg of plasmid containing the long isoform nucleic sequence of human, pig, or mouse LepR (fused at their C-terminal to the GFP), and 30 µL of lipofectamine 2000 reagent. The growth media was replaced 5 h after the transfection. Two days later, cells were collected with accutase and blocked in cold PBS-BSA 1% (*w*/*v*) at 2 × 10^6^ cells/mL for at least 30 min. Dilutions and washes were performed in the same buffer. Cells were then split into a 96-well conical bottom plate, pelleted by centrifugation (1 min at 750× *g*), and incubated for 30 min on ice with either an anti-LepR Nanofitin (1 µM, 100 µL/well) or an irrelevant Nanofitin (Irr-NF; 1 µM, 100 µL/well) or PBS-BSA 1% (*w*/*v*). Following that step, HEK293 cells were washed 3 times with 200 µL of PBS by centrifugation (1 min at 750× *g*), and cells previously incubated with Nanofitins were resuspended with the anti-His-tag antibody for 45 min on ice, whereas cells that were incubated with PBS-BSA 1% (*w*/*v*) were resuspended either with the anti-His-tag antibody (0.25 µg/mL, 100 µL/well) used as negative controls, or the anti-LepR antibody (0.3 µg/mL, 100 µL/well) used as a positive control. Both control antibodies are conjugated with Alexa-Fluor 647 for detection. Then, 3 washes were performed for all conditions and cells were resuspended into 200 µL of cold PBS-BSA 1% at a final concentration of 10^5^ cells/mL until analysis. The binding was recorded on viable single cells and on the allophycocyanine (APC) channel using a CytoFLex Flow Cytometer (Beckman Coulter, Brea, CA, USA), and data were analyzed with the CytExpert Sofware (Beckman Coulter).

### 2.5. Preparation for Ex Vivo Experiments

#### 2.5.1. Recovery and Transport of Porcine Intestinal Tissue

Directly after the animal slaughter, fresh porcine mid-jejunum was recovered from a 6-month-old Swiss noble pig (slaughter house of Loëx, Bernex, Switzerland) [35]. Tissue was rinsed with ultrapure water and immersed into oxygenated (95% O_2_ and 5% CO_2_, PanGas AG, Dagmersellen, Switzerland) cold krebs-bicarbonate ringer buffer (KBR, 120 mM NaCl, 20 mM NaHCO_3_, 11 mM glucose, 5.5 mM KCl, 2.5 mM CaCl_2_, 1.2 mM MgCl_2_, 1.2 mM NaH_2_PO_4_, at pH 7.4). The tissue was kept under oxygenation until the beginning of the experiment.

#### 2.5.2. Validation of the LepR Expression on Porcine Intestinal Tissue by Western Blot

To generate porcine intestinal tissue lysate for the evaluation of the LepR expression by Western blot, a small part of the porcine jejunum segment was crushed with liquid nitrogen, recovered in Radioimmunoprecipitation assay buffer (RIPA, 1 mM EDTA, 0.5 mM EGTA, 1% Triton X-100, 0.1% sodium deoxycholate, 0.1% SDS, 140 mM NaCl) supplemented with 1% of protease and phosphatase inhibitor cocktails then sonicated (3 times: 10 s, 5 cycles, 10%). Supernatant was recovered after 10 min of centrifugation (10,000 rpm, 4 °C) and proteins concentration was determined using a BCA protein assay kit. Then, 15 µg of porcine jejunum lysate, HepG2 cell lysate used as a positive control, HEK293 cell lysate used as negative controls, and a prestained protein ladder were separated on a 12% SDS-PAGE gel and transferred overnight at 4 °C to a nitrocellulose membrane (Cytiva). Blocking of the membrane during 4 h at RT was realized in TBST-BSA 3% (*w*/*v*) and followed by an incubation of 2 h at RT with a rabbit monoclonal antibody diluted 500 times, recognizing the long isoform of the LepR. After 3 washes of 10 min in TBST, an incubation of 2 h was carried out with a secondary anti-rabbit antibody diluted 10,000 times. The membrane was then washed 4 times and immunoreactivity was detected using chemiluminescence on the Odyssey imaging system (LiCor Biotechnology, Lincoln, NE, USA).

#### 2.5.3. Labeling of Nanofitins for Ex Vivo Transport Experiments

Anti-LepR Nanofitin and the irrelevant Nanofitin quantification for the ex vivo assays was performed via direct measurement of the fluorescence coming from the fluorescein conjugated Nanofitins. A regioselective conjugation of maleimide–fluorescein was performed on a unique engineered cysteine at the N-terminus of the Nanofitins. The modified Nanofitins were expressed from plasmid constructs, as described in Section 2.3, and dialyzed in Hepes-NaCl buffer (20 mM Hepes, 150 mM NaCl, pH 7.4) with SnakeSkin™ Dialysis Tubing 3.5K MWCO (Thermo Fischer Scientific). For the dimers, the single cysteine was added at the N-terminus of the constructs, i.e., at the N-terminus of the of the anti-LepR 1-F08 or the irrelevant Nanofitin domain.

Nanofitins were diluted at 5 mg/mL in Hepes-NaCl and reduced overnight at 4 °C with TCEP-HCl (Thermo Fischer Scientific) at a final concentration of 1 mM. The following day, N-(5-Fluoresceinyl)-maleimide was solubilized in dimethylsulfoxyde (DMSO) and added 10 times in excess to the reduced proteins. Solutions were left 2 h under stirring and argon at RT. Then, free dye was removed using a PD10 desalting column (GE Healthcare, Chicago, IL, USA), equilibrated with Hepes-NaCl, and the fluorescently labeled Nanofitins were concentrated until at least 10 mg/mL with 3 kDa protein concentrators (Thermo Fischer Scientific). Finally, protein concentrations were determined with a BCA Protein Assay Kit and their binding activity was validated by ELISA.

#### 2.5.4. Calibration of the Ussing Chambers and Porcine Intestinal Tissue Preparation

Anti-LepR Nanofitinstransport across porcine intestinal tissue was evaluated ex vivo in the Ussing chambers system [35]. To calibrate the system, two pairs of Ag/AgCl-electrodes were first inserted into tips filled with a mixture of 3% agar in KCl 3 M and connected to each chamber. Then, mucosal (donor) and serosal (receiver) compartments of the Ussing chambers were filled with 7 mL of preheated KBR (38 °C), and differences of voltage between the two pairs of electrodes, as well as the electrical resistance of the buffer, were removed. During this step, a circulating water bath ED-5 (Julabo GmbH, Seelbach, Germany) was used to regulate the temperature of the chambers (porcine body temperature: 38 °C). The system was also under constant oxygenation with a mixture of 95% O_2_ and 5% CO_2_ to keep the tissue viable and avoid unstirred layer formation.

At this stage, the tissues were mounted into the sliders. The intestine was first opened along the mesenteric border to remove the serosa and tunica muscularis via blunt dissection with fine scissors and tweezers. Then, segments of approximatively 1.5 cm^2^ were cut off, while being careful to avoid areas with Peyer’s Patches, and each of them was mounted on a slider (Physiologic instruments, San Diego, CA, USA) with an exposed surface area of 1.26 cm^2^. Then, the KBR buffer was removed, and the sliders were inserted between the donor and receiver compartments. After 30 min of equilibration of the tissues with preheated KBR, chambers were emptied again, and the buffer of the receiver compartments was replaced with fresh KBR to prevent a possible impact of endogenous material released during the equilibration. All segments were set within 60 min following the animal death into the Ussing chambers system. It was later coupled to a VCC MC6 Multichannel Voltage–Current Clamp (Physiologic Instruments) to monitor the setup integrity by continuous measurements of the transepithelial electrical resistance (TEER, in Ω·cm^2^).

### 2.6. Ex Vivo Transport Evaluation of Nanofitins Targeting the LepR

To evaluate the anti-LepR Nanofitins transport across porcine intestinal tissue, they were diluted in KBR at 90 µM. The irrelevant Nanofitin was diluted at the same concentration. With the limit of quantification by fluorescence in KBR being 18 ng/mL, the used concentration allowed for a quantification of the fluorescent proteins added into the mucosal compartments as low as 0.0019%. Then, each donor compartment was filled with 7 mL of the Nanofitin solutions. For competition assays, a 9-fold excess of unlabeled anti-LepR Nanofitin 1-F08 (800 µM) was also added to the mucosal compartments. After 120 min of experiment (maximal viability duration of the intestinal tissue in Ussing chambers [6]), tissues exhibiting a TEER value below 15 Ω·cm^2^ were omitted from the dataset as this threshold reliably indicates a loss of integrity [36]. Final donor and receiver samples were recovered from the compartments of the other chambers.

Although the common analytical method employed to quantify small molecules following transport experiments lean on UHPLC-MS/MS [35], we chose fluorescence to measure the transported Nanofitins given the higher sensitivity of this method for these proteins. To that end, the Nanofitins in the donor compartment at the beginning of the experiment (initial) and after 120 min (final) were diluted 500 times in KBR to fit into the detection range fixed by a calibration curve performed in KBR with solutions ranging from 0.8 to 900 nM (each dilution was performed in triplicate). The buffer was used as a blank. In contrast to the Nanofitins in the donor solution, the Nanofitins in the receiver compartment were used undiluted (each solution was tested in triplicate). A total of 200 µL of all the samples were then added into a black Nunc MicroWell 96-Well Optical-Bottom plate (Thermo Fischer Scientific). Fluorescence (λex: 490 nm; λem: 525 nm) was measured by a CLARIOstar Plus plate reader (BMG Labtech, Ortenberg, Germany).

In order to determine the percentages of transported Nanofitins (anti-LepR Nanofitins or irrelevant Nanofitin) in the receiver compartments at the end of the experiments, the following equations were used with the masses determined by the fluorescence calibration curve and considering the used dilutions:(1)$$Transported\;Nanofitin\;(\%\;initial\;donor)=\frac{mass\;transported\;in\;the\;receiver\;after\;120\;min}{mass\;in\;the\;donor\;compartment\;at\;the\;onset}\times 100,$$
(2)$$Final\;donor\;(\%\;initial\;donor)=\frac{mass\;in\;the\;donor\;compartment\;after\;120\;min}{mass\;in\;the\;donor\;compartment\;at\;the\;onset}\times 100$$

### 2.7. Statistical Analysis

All statistical analyses were performed using the GraphPad Prism 7.0 software. Data are presented as the mean ± standard deviation (SD). The results obtained were evaluated with the Student’s *t*-test. The significance level was fixed at α = 0.05.

## 3. Results

### 3.1. Characterization of Nanofitins Targeting the LepR

#### 3.1.1. Selection of Nanofitins Targeting the LepR for Ex Vivo Transport Experiments

To identify Nanofitins binding to the extracellular domain of the hLepR, a combinatorial library of variants was challenged through four rounds of ribosome display using the rhLepR as the target. Following the final selection round, the enriched library was subcloned and transformed in *E. coli*, and 95 single clones were isolated for a preliminary screening fused with acGFP against the rhLepR by ELISA. Clones were ranked based on their binding signal ratio on the rhLepR versus non-specific binding signal in the absence of the target (Appendix A). The enrichment of specific hits led to the isolation of 53 unique validated clones exhibiting a specific signal on rhLepR at least twice superior to the non-specific signal (Appendix A). These clones were categorized into four families based on similarities in amino acid patterns within their binding site.

Subsequently, a representative panel of 20 hits, selected based on their binding signal-to-noise ratio and sequence diversity within and between families, was produced in the same format for further characterization. Various binding profiles were identified among these clones. A competition assay with the natural ligand, the leptin, also revealed that all clones from family 1 displayed a binding ratio close to 1, demonstrating a specific interaction with rhLepR on an epitope distinct from that of leptin. Conversely, families 2, 3, and 4 exhibited binding ratios of 2 or higher, suggesting binding to an epitope that overlapped, at least partially, with the leptin binding site (Figure 1A). Finally, cross-reactivity experiments performed with rmLepR showed that clones from family 1 present a similar binding signal to the mouse and human targets, whereas the others have a higher binding signal than the mouse receptor (Figure 1B). These results demonstrate that the identified Nanofitins target at least two distinct epitopes on the LepR.

We focused the next stages of the study on anti-LepR Nanofitins 1-F08 and 1-D07 from family 1. This family comprised Nanofitin candidates displaying two essential features from the target product profile: non-competition with the binding of the natural ligand to prevent any interference with natural physiological processes [37,38]; and binding both the murine and human forms of the receptor to facilitate translatability during preclinical evaluations. Within family 1, 1-F08 and 1-D07 were chosen due to their differing ELISA response levels, suggesting distinct affinities.

#### 3.1.2. Binding Characterization of Selected Nanofitins

Recognizing the significance of affinity in BBB studies for modulating active transport across cellular barriers [39], we conducted a more in-depth characterization of the binding affinity of both 1-F08 and 1-D07 in a format without acGFP (Figure 2).

The EC_50_ values obtained in ELISA indicated a higher relative affinity for 1-F08 (0.29 nM) compared to 1-D07 (35.70 nM) (Figure 2A,B). Moreover, we investigated the binding kinetic parameters of these two anti-LepR Nanofitins by BLI. The sensorgrams confirmed the high affinity of 1-F08 for the target (K_D_ = 9.56 nM) and emphasized the strong contribution of the on-rate to its affinity with rapid association kinetics (k_on_ = 1.63 × 10^6^ M^−1^s^−1)^ (Figure 2A,C). Conversely, 1-D07 exhibited a lower affinity for the target (K_D_ = 830 nM) compared to 1-F08. Notably, 1-D07 demonstrated a slower association rate (k_on_ = 4.21 × 10^5^ M^−1^s^−1^) than 1-F08, as well as a faster dissociation from the target (k_off_ = 3.49 × 10^−1^ s^−1^ for 1-D07 and k_off_ = 1.56 × 10^−2^ s^−1^ for 1-F08), leading to a distinctive square waveform profile.

#### 3.1.3. In Vitro Evaluation of Nanofitins Cross-Reactivity on Transfected HEK293 Cells

As shown in Figure 1B, cross-reactivity was detected on human and murine isolated receptors. This was further supported by assessing the specific binding of Nanofitins on cell models expressing the LepR from different species. To that end, HEK293 cells were transiently transfected with plasmids encoding for the long isoform of the LepR from human, mouse, and swine. This approach aimed to encompass all relevant models likely to be used for the preclinical evaluation of the transport abilities of the anti-LepR Nanofitin-based shuttle. Transfection efficiency was monitored by measuring cells fluorescence since each LepR was fused with GFP to facilitate cell selection. The expression of receptors at the cell membranes was confirmed by the specific binding of a labeled anti-LepR antibody on transfected cells compared to non-transfected cells (Appendix A).

The binding of 1-F08 and 1-D07 was tested at 1 µM by flow cytometry and was normalized with the anti-LepR antibody binding signal on cells (Figure 3). The anti-His antibody alone and an irrelevant Nanofitin were also tested to establish the correlation between signal positivity and the presence of specific anti-LepR Nanofitins (Appendix A). Furthermore, the absence of Nanofitin binding on non-transfected HEK293 cells was confirmed in each assay.

The data presented in Figure 3 revealed that both anti-LepR Nanofitins bind to the hLepR, as demonstrated by their significantly higher signal on cells compared to the negative control. Their level of positivity aligns with their respective affinities determined in Figure 2. Additionally, 1-F08 recognizes the membrane-bound porcine LepR and the mouse LepR, confirming its cross-reactivity with these targets. We observed minimal to no cross-reactivity with the anti-LepR Nanofitin 1-D07, attributed to its lower affinity in conjunction with higher background signals on these cell models.

### 3.2. Experimental Setup Validation for Ex Vivo Studies

#### 3.2.1. LepR Expression in Porcine Jejunum Lysate

Since the jejunum exhibits the most active carrier-mediated transport in the intestine [40], our focus for the ex vivo studies was directed toward this particular segment of the GI tract. To investigate the expression of LepR in this tissue, we conducted an immunoblot analysis. The immunoblot revealed a band at the expected molecular weight (130 kDa) in the porcine jejunum lysate, which was absent in the negative control (HEK293 cell lysate). The same band, representing the long isoform of the receptor, was identified in the positive control (HepG2 lysate, Appendix A). These results validate the expression of LepR in the porcine jejunum and underscore the relevance of using an ex vivo model of porcine origin to evaluate the intestinal transport of Nanofitins targeting this receptor.

#### 3.2.2. Nanofitins Fluorescently Labeled

The anti-LepR Nanofitins and the irrelevant Nanofitin were fluorescently labeled using N-(5-Fluoresceinyl)-maleimide. We determined the EC_50_ of the labeled constructs by ELISA on immobilized rhLepR. No discernible impact of the labeling was observed on the binding activities of the anti-LepR Nanofitins. The EC_50_ values obtained (4.23 × 10^−10^ M and 2.89 × 10^−8^ M for fluorescent 1-F08 and 1-D07, respectively, Appendix A) fell within a similar range to those determined for the unlabeled proteins (2.94 × 10^−10^ and 3.57 × 10^−8^ M for 1-F08 and 1-D07, respectively, Figure 2).

### 3.3. Investigation of Receptor-Mediated Transport across the Intestinal Barrier

#### 3.3.1. Specific Transintestinal Shuttle Activity of Non-Competitive Anti-LepR Nanofitins

The ability of fluorescein-labeled 1-F08 and 1-D07 to cross the intestinal barrier was assessed in comparison to an irrelevant Nanofitin used as a negative control (Figure 4).

Continuous monitoring by TEER measurements yielded a mean TEER of 45 ± 16 Ω·cm^2^ (n = 53) at the conclusion of the incubation period of labeled Nanofitins in the donor (mucosal) compartments of the Ussing chambers. This value closely resembles the reported ex vivo measurements for porcine (42 ± 14 Ω·cm^2^) [35] and human intestines (34 ± 12 Ω·cm^2^) [41], confirming the integrity of each considered intestinal segment. The percentages of labeled Nanofitins found in the receiver compartments were calculated relative to the quantity of material applied in the donor compartments, determined using the calibration curve equation (an example of such a calibration curve is provided in Appendix A).

The anti-LepR Nanofitin displaying the highest passage across the intestinal barrier is 1-F08 (0.16% of the initial donor or 11.59 ± 4.40 µg) demonstrated a 3.2-fold increase compared to the calculated passage for the irrelevant NF (0.05% of the initial donor or 4.02 ± 0.98 µg) (Figure 4A). Similarly, the passage percentage of 1-D07 is significantly higher than the negative control, presenting a 2.6-fold difference (0.13% of the initial donor, or 10.51 ± 3.66 µg). These data demonstrate that LepR-binding Nanofitins exhibit a greater capability to cross the intestinal barrier compared to the irrelevant Nanofitin, suggesting a potential role of LepR in their transport. Notably, due to its superior ex vivo transport efficacy and higher in vitro affinity for the LepR, 1-F08—belonging to the same family as 1-D07—was chosen for further investigation of the receptor-mediated transport across the intestinal barrier.

To further validate the potential specific interaction driving the transportation of 1-F08 across the intestinal barrier, we conducted a competitive assay. This involved employing labeled 1-F08 along with an excess of either unlabeled 1-F08 or an irrelevant Nanofitin in the donor compartment, in comparison to labeled 1-F08 used solely (for normalization). When an excess of unlabeled irrelevant Nanofitin was introduced, it resulted in a modest 28% reduction in the transported amount of labeled 1-F08 in the receiver compartments. However, the addition of unlabeled 1-F08 led to a notably higher reduction, showing a 58% decrease (Figure 4B). These results suggest a specific passage mechanism for 1-F08, likely triggered by its binding to the LepR.

The ability of 1-F08 to transport a cargo protein was also evaluated by fusing the anti-LepR Nanofitin with a second Nanofitin (NF2) specific to HSA [28], serving as a model protein for potential carriage by the shuttle. We assessed the binding activity of 1-F08-NF2 on the rhLepR and HSA by ELISA. Similar EC_50_ values were obtained on the rhLepR for 1-F08, whether tested as a monomer or as a dimeric construct (Appendix A), indicating consistent binding activity on the rhLepR. Furthermore, the low EC_50_ obtained for HSA aligns with the reported high affinity of this Nanofitin for HSA [28]. Thus, both Nanofitins conserved their respective binding activities in genetic fusion. Following this validation, the dimeric construct was labeled with fluorescein, and its binding conservation was confirmed by ELISA.

The transport of 1-F08-NF2 was evaluated ex vivo against a labeled dimeric construct, wherein 1-F08 is substituted by an irrelevant Nanofitin (Irr NF-NF2). The results reveal that the passage percentage of 1-F08-NF2 (0.17% of the initial donor, or 26.24 ± 4.91 µg) is significantly higher compared to the negative control Irr NF-NF2 (0.06% of the initial donor, 11.33 ± 3.01 µg), showing a 2.7-fold difference. Moreover, similar crossing efficiency was observed for 1-F08 alone and 1-F08-NF2 (0.16% and 0.17% of the initial donors, respectively) (Figure 4C). These data demonstrate that the molecular shuttle 1-F08 can promote the transport of a cargo (NF2) across the intestinal barrier as efficiently as in the absence of a cargo.

#### 3.3.2. Nanofitin Activity Conservation after Mucosal Transport

In order to assess the binding activity of the monomeric and dimeric constructs subsequent to their transit through porcine jejunum, ex vivo samples of 1-F08 and 1-F08-NF2, retrieved from receiver compartments after 120 min, were tested by ELISA (Figure 5). These samples were compared to those obtained from their respective initial donors, which were diluted in KBR at the same concentration. The specificity of the signals was validated by the low absorbance observed in wells without target for all samples.

The results demonstrate that both 1-F08 and the dimeric construct maintained their capability to bind the rhLepR target after crossing the intestinal epithelium. Additionally, this validates the structural integrity of a substantial portion of the protein fusion 1-F08-NF2, as the RGS-His detection tag was genetically appended to the C-terminus of NF2. Furthermore, the Nanofitin cargo (NF2) within the dimeric construct exhibits binding to its target, HSA, suggesting the shuttle’s capacity to transport an active protein. Nevertheless, the binding signals of 1-F08 and 1-F08-NF2 receiver samples on rhLepR exhibit reductions of 33% and 42%, respectively, in comparison to the binding signals obtained with their initial donors at equivalent concentrations. Similarly, the binding signals of the dimeric construct from receiver samples, assessed on immobilized HSA, exhibit a reduction by half compared to those obtained from the initial donor. Hence, further investigations are warranted to elucidate the underlying factors contributing to the observed differences in binding signals. These variations might arise from protein derivatization, potential partial denaturation, or artifacts associated with the in vitro and ex vivo methods.

## 4. Discussion

The oral administration of systemically bioavailable biologics presents considerable challenges due to their vulnerability to the harsh conditions of the GI tract and their limited absorption through the intestinal barrier [1,42,43]. Overcoming these obstacles requires a strategy that facilitates active and specific mucosal transport while withstanding chemical and proteolytic stresses from the buccal cavity to the intestine. Additionally, it is crucial for the carrier to exhibit controlled conjugation or fusion compatibility [30,31,44], enabling therapeutic activity and precise modulation of clearance once in the bloodstream [28]. In this study, we demonstrate the potential of Nanofitins, engineered from a 7 kDa hyperstable protein scaffold [25,26,45], to selectively target the LepR [19] and trigger the receptor-mediated transport of protein payloads from the gut lumen to the bloodstream, as demonstrated using an ex vivo model.

The anti-LepR Nanofitin target product profile was meticulously designed during the early stages of the project, considering its pivotal role in transitioning from wet lab experiments to preclinical studies and subsequent human use. Key criteria were given priority as follows: selecting Nanofitins that recognize the human LepR with non-overlapping epitopes compared to the natural ligand emerged as a primary focus. This strategy aimed to bolster efficiency and safety by mitigating risks associated with competition with endogenous ligands present at physiological concentrations, thereby reducing potential interference with normal biological functions [24,46,47]. Secondly, substantial emphasis was placed on identifying a Nanofitin capable of cross-reaction with mouse and pig species. This cross-reactivity facilitates the use of ex vivo and in vivo models during preclinical stages while upholding a high level of translatability. Furthermore, although preclinical efficacy studies may occasionally use humanized animals [48,49], they are commonly based on murine pathological models. Thus, the inclusion of Nanofitins exhibiting inter-species cross-reactivity obviates the necessity for numerous animal models specific to each active pharmaceutical ingredient, streamlining effective transport across the intestinal barrier.

To assess the transport capability of anti-LepR Nanofitins, we established a setup using porcine intestinal sections within Ussing chambers [35]. This approach provided a controlled experimental environment, enabling us to focus specifically on studying the transport process without the influence of systemic factors encountered in vivo. Beyond exploring the transport mechanism, this ex vivo model offered an opportunity to refine experimental techniques, optimize protocols, and gather valuable insights before transitioning to in vivo studies. Consequently, this strategy empowered informed decision-making, elevating the likelihood of success while reducing dependence on animal subjects [50,51]. In addition to ethical considerations, prior studies have highlighted a strong resemblance between porcine and human intestinal tissues in macroscopic and microscopic characteristics, establishing the porcine model as a reliable predictor of drug absorption in humans [52,53]. While acknowledging that reducing an entire digestive organ to a tissue section in a laboratory setting only partially mirrors the physiological diversity along the intestinal tract, we focused on the jejunal region due to its relevance to this study. Notably, the jejunum exhibits the highest degree of carrier-mediated transport in the gut, making it particularly suitable for our investigations [40,54]. Moreover, our validation of the presence of the long isoform of the LepR—responsible for leptin’s intestinal transcytosis [19]—in porcine jejunal tissue (Appendix A) reinforced the suitability of our ex vivo model. Thus, this ex vivo system emerges as a reliable mimic of the human intestine, pivotal for preclinical investigations and for evaluating the efficacy of the anti-LepR Nanofitins. This study marks a significant advancement by utilizing the porcine intestinal model within Ussing chambers to assess the transport of biologics, expanding from its conventional usage restricted to small compounds [35]. It represents a noteworthy progression in comprehending gastrointestinal transport mechanisms through an ex vivo model and underscores the pioneering nature of this research, potentially opening new avenues in the field of oral drug delivery and therapeutic applications.

To initiate our study, we conducted a successful screening of the Nanofitins library, leading to the identification of four major families. Notably, Nanofitins from family 1 exhibited binding to the LepR that remained unaffected by the presence of human leptin, distinguishing them from Nanofitins from the other families (Figure 1A). This competitive behavior coincided with the results of cross-reactivity screening, wherein family 1 Nanofitins displayed ELISA signals independent of the target host (Figure 1B). As an outcome, we selected 1-F08 as the primary candidate for the proof-of-concept study. Acknowledging the limited understanding of the optimal parameters required to trigger active passage and enhance intestinal permeability of biologics [12,39], we included its homologue 1-D07, possessing lower affinity (a two-log difference in EC_50_ and K_D_ values (Figure 2)) for characterization alongside 1-F08.

We hypothesized that this affinity disparity significantly impacts the binding to cell surface receptors, as evidenced by flow cytometry analysis following transient transfection of human, mouse, and pig LepR in HEK293 cells (Figure 3). While 1-F08 exhibited robust labeling of LepR-positive cells in all conditions, 1-D07 yielded modest signals with the human target, and the cross-reactivity could not be confirmed in a cellular context. However, both candidates showed promise in assessing our strategy regarding the ability of Nanofitins targeting LepR to cross the intestinal barrier using the ex vivo porcine model.

Transport experiments conducted using the Ussing chamber system revealed that 1-F08 and 1-D07 exhibited a 3.6-fold- and 2.5-fold-higher crossing ability, respectively, from the mucosal to the serosal compartment compared to the irrelevant Nanofitin (Figure 4A). Consistent with recent studies on FcRn-mediated transcytosis [39], we observed a correlation between increased affinity for the target receptor and enhanced transport across the intestinal barrier. Additionally, competition experiments demonstrated a significant hindrance, with a 58% decrease in the intestinal transport of labeled 1-F08 in the presence of an excess of unlabeled 1-F08, while only a 26% decrease was observed with the unlabeled irrelevant Nanofitin (Figure 4B). These findings suggest the saturation of the LepR epitope targeted by unlabeled 1-F08, impeding the interaction between the fluorescent protein and the receptor. Then, we tested a construction composed of 1-F08 with NF2, an anti-HSA Nanofitin [28], to illustrate the transportation of other biological molecules, in that case another protein fused genetically to 1-F08 (Figure 4C). The results strongly support the notion that anti-LepR Nanofitins from family 1 can serve as a shuttle for the transportation of other biologics, as demonstrated by the specific transport of the anti-HSA Nanofitin, NF2. These results further validate the concept that 1-F08 uses a cargo-size-independent mechanism [55], addressing the challenge of transporting diverse biological molecules, such as a protein, across the intestinal mucosa.

Using the ex vivo model samples, we demonstrated the effective transport of both monomers and dimers of Nanofitins including the 1-F08 moiety, while retaining their specific binding activity. This was exemplified by the successful transport of the anti-LepR 1-F08 and anti-LepR/anti-HSA 1-F08-NF2 constructs (Figure 5). Despite acknowledging a partial reduction in the binding signals of the constructs in the serosal compartments, these results affirm the integrity of the multimeric assembly detected by ELISA on the rhLepR. Notably, the anti-LepR activity was elicited by the N-terminus moiety, while the detection tag was located at the C-terminus of the NF2. These data show the preserved binding activity of the constructs and highlight the significant potential of 1-F08 as a gut-to-blood shuttle for delivering active molecules with therapeutic purposes. However, it is important to consider the challenges posed by gastric and intestinal fluids [42], as well as the consecutive requirements for stability in future constructs. These considerations become particularly crucial for successful oral delivery of therapeutic molecules with maintained efficacy. Robust moieties, such as small molecules, peptides, or Nanofitin-based compounds, may offer improved resistance to the demanding conditions of the GI tract. Various strategies [4,6] can be explored to address the challenging conditions of the GI tract, warranting consideration for the forthcoming preclinical experiments subsequent to this proof-of-concept study.

Translating the findings of our study into in vivo drug bioavailability and subsequent therapeutic effect necessitates further investigation. Our initial results demonstrate the transport capabilities of anti-LepR 1-F08, capable of transporting itself and another Nanofitin domain at similar rates (NF2). This is exemplified by the observation that the dimeric construct, with a molecular weight twice that of the monomer, appeared in double the mass of the monomer in the receiver compartment. We envision the potential addition of more moieties, especially at the C-terminus of 1-F08, in future biotherapeutics. Additionally, the 1.26 cm^2^ of jejunal surface exposed in an Ussing chamber represents only a fraction of the available surface in the human jejunum, estimated at approximately one 1000th to one 10,000th depending on diameter and length [56]. Assuming linear rate conversion, the 26.24 µg of 1-F08-NF2 available in the serosal compartment after 2 h could translate to a dosage range from hundreds of µg/kg to single-digit mg/kg after each administration. These expected blood titers underscore the potential of the anti-LepR shuttle for real therapeutic applications, enhanced by the ability to fine-tune blood accumulation through the use of an anti-HSA moiety [28] in the final product. Furthermore, Nanofitin-based molecules are notably smaller than monoclonal antibodies. As a consequence, an equivalent molar dose of the pharmaceutical ingredient falls within the standard range of circulating monoclonal antibodies, starting from 1 mg/kg [57,58]. These prospects will undergo further assessment, considering numerous factors influencing final serum bioavailability, including: (i) actual exposure to the intestine, accounting for a transit time through the human intestine of up to 6 h [56]; (ii) expression levels of LepR across the human gut tract, evidenced at the RNA level from the stomach to the rectum, not limited to the jejunum [59,60,61]; (iii) receptor turnover rate, crucial for optimizing dosing regimens and determining the administration frequency [62]. These considerations are paramount in ensuring the successful and effective utilization of biologics in therapeutic applications.

## 5. Conclusions

In summary, this ex vivo proof-of-concept study has yielded compelling insights supporting several critical observations. First, it demonstrates the efficacy of LepR as a target capable of initiating a specific transport mechanism, showcasing the potential for leveraging this receptor in designing innovative oral-to-systemic delivery strategies. Second, the study conclusively demonstrates that a Nanofitin targeting a distinct epitope from the natural ligand of the LepR can efficiently traverse the intestinal barrier via a translocation mechanism, highlighting the promise of anti-LepR Nanofitins as carriers for overcoming biological barriers. Moreover, this study has established the potential of 1-F08 as a shuttle by enhancing the permeability of other biologically active molecules, aligning with the growing demand for efficient drug delivery systems in the field of biologics.

With these impactful findings, our research can embark on the next phase of advancement. This forthcoming stage will involve a comprehensive evaluation of the in vivo transport dynamics of 1-F08, engineered to incorporate a stable, biologically active cargo in rodent models. This crucial step bridges the gap between ex vivo insights and real-world applications, offering valuable insights into the construct’s behavior within the complexities of living organisms. It marks a critical steppingstone toward translating our innovative approach into potential clinical applications.

## 6. Patents

The innovative findings presented in this manuscript have led to the development of a patent entitled “Polypeptides as Carrier for Intestinal Barrier Crossing” (application number: EP21306709.3; file date: 5 December 2021). The patent, filed based on the novel discoveries reported herein, covers the unique application of Nanofitins as carriers facilitating the transport of therapeutic compounds across the intestinal barrier. The inventive approach outlined in this manuscript forms the basis for the claims made in the patent application, highlighting the potential for significant advancements in drug delivery systems.

## Figures and Tables

**Figure 1 pharmaceutics-16-00116-f001:**
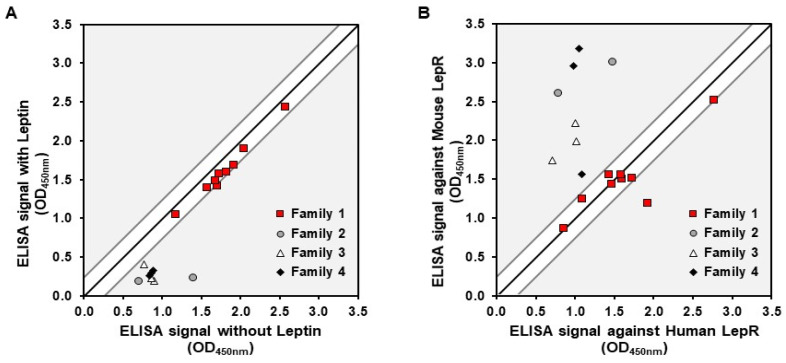
Characterization of Nanofitins targeting the LepR. (**A**) Signals obtained in a competition ELISA with Nanofitins tested on immobilized rhLepR in the presence or absence of leptin. (**B**) Signals obtained with a cross-reactivity ELISA with Nanofitins tested on immobilized rhLepR or rmLepR. Signal ratio of 1:1 is indicated by a black line, with a delimitation of ±0.25 absorbance units.

**Figure 2 pharmaceutics-16-00116-f002:**
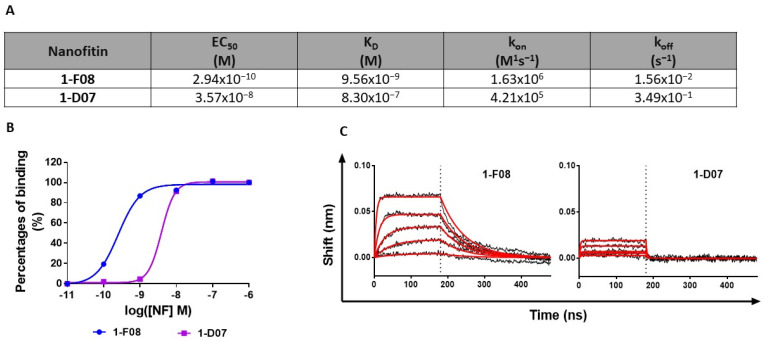
Characterization of the interaction of 1-F08 and 1-D07 with rhLepR. (**A**) Summary of binding parameters (EC_50_, K_D_, k_on_, and k_off_) of 1-F08 and 1-D07 (n = 1). (**B**) ELISA binding curves of 1-F08 and 1-D07 on the immobilized rhLepR (n = 1). Signal normalization was achieved by setting the highest binding signal obtained for each protein as 100%. (**C**) BLI binding profiles (association for 180 s followed by dissociation for 300 s) of 1-F08 and 1-D07 (n = 1). The concentrations represented start from 111 nM for 1-F08 and 333 nM for 1-D07 (followed by successive one-third dilutions). Raw data are represented in black sensorgrams, whereas fitted sensorgrams are in red.

**Figure 3 pharmaceutics-16-00116-f003:**
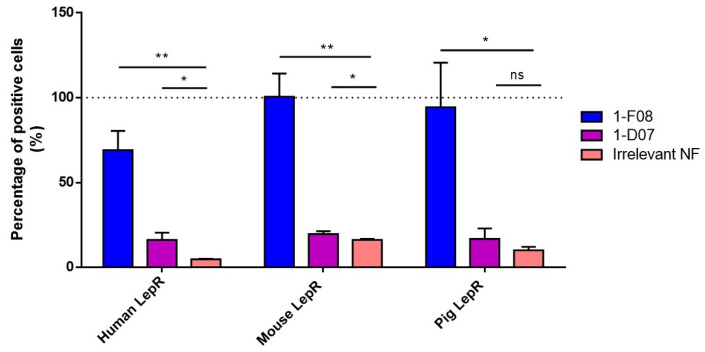
Binding of 1-F08 (n = 4) and 1-D07 (n = 4) on HEK293 cells transfected with the human, mouse, and pig LepR, analyzed by flow cytometry. The binding signals were normalized with the anti-LepR antibody binding signal (100%, represented by the dashed line) and expressed as a percentage of positive cells. An irrelevant Nanofitin (n = 2) was used as a negative control. Results are expressed as mean values ± SD (unpaired Student’s *t*-test; ns, non-significant (*p* > 0.05); * *p* < 0.05, ** *p* < 0.01).

**Figure 4 pharmaceutics-16-00116-f004:**
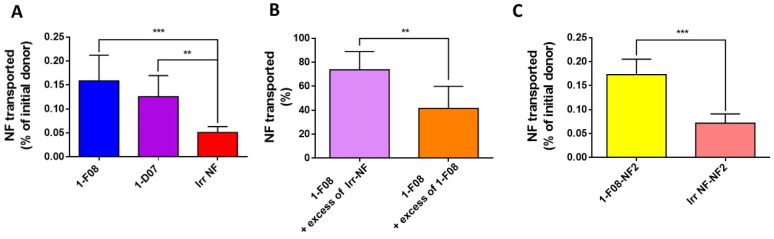
Ex vivo transport experiments across porcine intestinal tissue performed with (**A**) labeled anti-LepR Nanofitins 1-F08 (n = 10), 1-D07 (n = 4), and an irrelevant Nanofitin (n = 6) at 90 µM; (**B**) labeled 1-F08 at 90 µM in the presence of a 9-fold excess of unlabeled 1-F08 (1-F08 + excess of 1-F08, n = 6) or unlabeled irrelevant Nanofitin (1-F08 + excess of Irr-NF, n = 8); or (**C**) labeled 1-F08-NF2 (n = 4) and labeled irrelevant NF-NF2 (Irr NF-NF2, n = 5) tested at 90 µM. The masses used to calculate the percentages of initial donor transported were determined by fluorescence. The percentages of Nanofitin transported (**B**) were normalized with the percentage of labeled 1-F08 transported in the absence of an excess of unlabeled Nanofitin. Results are expressed as mean values ± SD (unpaired Student’s *t*-test; ** *p* < 0.01, *** *p* < 0.001).

**Figure 5 pharmaceutics-16-00116-f005:**
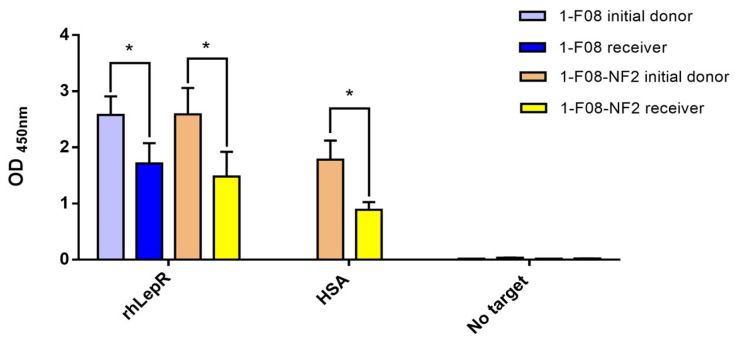
Binding activity of ex vivo samples evaluated on immobilized rhLepR or HSA. The initial donor samples of 1-F08 (n = 4) and 1-F08-NF2 (n = 3) were diluted at the same concentrations as their respective receiver samples. Specificity of Nanofitins binding was validated on wells without target. Results are expressed as mean values ± SD (unpaired Student’s *t*-test; * *p* < 0.05).

## Data Availability

Data are contained within the article.

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
