# Peer review of "Enhancing Oral Delivery of Biologics: A Non-Competitive and Cross-Reactive Anti-Leptin Receptor Nanofitin Demonstrates a Gut-Crossing Capacity in an Ex Vivo Porcine Intestinal Model"

_pharmaceutics, 2024, doi:10.3390/pharmaceutics16010116_

Round 1

Reviewer 1 Report

Comments and Suggestions for Authors

The manuscripts report the proof-of-concept of the use of Nanofitin as a tool for overcoming biological barriers for the oral administration of biopharmaceuticals. The originality and relevance of the work are considerable and the approach is new.

The manuscript is well written and well organized.

Very minor comments are:

- page 8, lines 359-361: the authors should indicate a reason for choosing fluorescence instead of UHPLC-MS/MS for quantification. Some more detail on analytical method validation could be added.

- page 8, lines 375-378: the authors should indicate the reason for choosing an experimental time of 120 min.

Author Response

Dear reviewer,

Thank you very much for taking the time to review our manuscript. Please find the detailed responses below and the corresponding revisions/corrections in track changes in the re-submitted file.

  • We appreciate your insightful comment. The decision to employ fluorescence for quantification of transported proteins, as opposed to the commonly used UHPLC-MS/MS method for small molecules, was driven by the higher sensitive detection of Nanofitins. We have incorporated this rationale into the manuscript to better explain our methodology.

  • We acknowledge your query regarding the rationale for choosing a 120-minute experimental duration. The 120-minute timeframe was selected based on considerations related to the physiological relevance of the experimental setup. This duration aligns with the approximate viability duration of the porcine intestinal tissue in our ex vivo transport study [6]. We have included this explanation in the manuscript to provide clarity on the choice of experimental duration. In more detail, previous studies were performed in our set-up to define the threshold transepithelial electrical resistance (TEER) value below which the tissue integrity was considered to be impaired or not viable as well as the maximum duration the experiment can be performed. It was concluded that tissues with a TEER below 15 W.cm2 were considered not to be viable or intact and 2 hours was the maximum duration for measurements in appropriate conditions.

Reviewer 2 Report

Comments and Suggestions for Authors

The manuscript submitted by Solene Masloh et al contains promising results concerning the use of Nanofitins as a transintestinal shuttle for biologically active molecules. One may guess a massive amount of experimental data behind what authors chose to present in this well-organized and well-written manuscript. I've been able to detect a handful of minor errors, marked in the attached, reviewed version. I would like to make a suggestion, based on Figs. 2 and 4. With more Nanofitin candidates (1-F08, 1-D07, ...) and the corresponding intestinal crossing rates available, one may expect the best shuttle to exhibit an "intermediate" binding affinity to rhLepR (i.e. the best binder may not necessarily be the best shuttle). A plot of the crossing activity against binding affinity may look like volcano-shaped, as first illustrated in Paul Sabatier's principle of heterogeneous catalysis. Its applicability is also well documented for enzymatic processes.

Author Response

Dear reviewer,

Thank you for your constructive feedback and insightful suggestion regarding the manuscript on Nanofitins as a transintestinal shuttle. We appreciate your positive assessment and diligent review of the document, including the marked minor errors, which we promptly addressed. Your suggestion to explore the relationship between Nanofitin binding affinity to rhLepR and their intestinal crossing rates is indeed interesting. We recognize the potential benefits of broadening our evaluation to encompass a wider array of Nanofitin candidates. In a forthcoming study, leveraging our analytical setup, we aim to create a comprehensive plot elucidating the relationship between crossing activity and binding affinity. Concurrently, we are developing in vitro models designed for higher throughput and reduced protein consumption. This approach aims to streamline the identification of the most diverse and promising hits. This extension will enable us to determine whether, as anticipated, the optimal shuttle typically exhibits an intermediate binding affinity.

Reviewer 3 Report

Comments and Suggestions for Authors

The research article titled “Enhancing oral delivery of biologics: a non-competitive and cross-reactive anti-leptin receptor Nanofitin demonstrates a gut-crossing capacity in an ex vivo porcine intestinal model” is interesting. This work demonstrated the potential of engineered Nanofitins, from a protein scaffold to selectively target the LepR and activate the receptor-mediated transport of protein payloads from the gut lumen to the bloodstream.

1.       For ex vivo transport studies, the authors indicate that “After 120 minutes of experiment, tissues with a TEER value lower than 15 Ω.cm2 were excluded from the data”. It is not clear whether the lowering of TEER values is due to the effect of Nanofitin solutions exposure. What is the impact of size and concentration of Nanofitins?

2.       While it is interesting to see the encouraging results from nanofitins evaluation executed via ex vivo studies, there is no indication of in vivo studies and how the nanofitins behave with respect to biological stability and drug permeability.

3.       Does the nanofitins transport/translocate across the intestine without change to their structural integrity? Has the mechanism of transport been well characterized? Could the authors illustrate the mechanism of receptor mediated nanofitins transport in a graphical way in this manuscript?

4.       It is also not clear if the nanofitins can only be conjugated to LepR or can they be conjugated with their respective targeted moieties to other well-expressed transporters/receptors on intestine?

Author Response

Dear reviewer,

Thank you very much for taking the time to review our manuscript. Please find the detailed responses below and the corresponding revisions/corrections in track changes in the re-submitted file.

Comment 1: Thank you for your insightful comment. Regarding the exclusion criterion based on TEER values in our ex vivo transport studies, we want to clarify that the decision to exclude tissues with a transepithelial electrical resistance (TEER) value lower than 15 Ω.cm2 after 120 minutes is not directly attributed to the exposure to Nanofitin solutions. The viability of porcine intestinal tissue in Ussing chamber setups is known to be approximately 120 minutes, and the threshold of 15 Ω.cm2 reliably indicates loss of tissue integrity (Arnold YE, Kalia YN. Using Ex Vivo Porcine Jejunum to Identify Membrane Transporter Substrates: A Screening Tool for Early—Stage Drug Development. Biomedicines. 2020 Sep 10;8(9):340). We have included this explanation in the manuscript to provide clarity on the choice of experimental duration and thresholds. Practically, previous studies were performed in our set-up to define the threshold TEER value below which the tissue integrity was considered to be impaired or not viable as well as the maximum duration the experiment can be performed. It was concluded that tissues with a TEER below 15 W.cm2 were considered not to be viable or intact and 2 hours was the maximum duration for measurements in appropriate conditions.

Moreover, to assess the impact of Nanofitin solutions exposure during our preliminary study, we performed viability tests on the tissue, both with and without Nanofitins, and found the viability duration to be approximately similar under these conditions. We appreciate your attention to detail and hope this clarification addresses your concerns.

Comment 2: Thank you for your thoughtful comment. We are currently conducting in vivo studies to investigate the behavior of Nanofitins concerning biological stability and drug permeability following this first article presenting our proof-of-concept study. These in vivo studies are a crucial component of our research, and we intend to provide comprehensive insights into the Nanofitins' performance in a physiological context. We anticipate sharing the outcomes of our in vivo studies in subsequent publications, which will provide valuable information to the overall understanding of the Nanofitins' efficacy and stability. We thank you for your patience and look forward to presenting the complete findings of our research.

Comment 3: Thank you for pointing out the mechanistic aspect of the NFs transport. We agree this would be great to understand more deeply how that works. In the ex vivo model, we were not able to decipher the exact mechanism used by NFs to be transported through the intestinal barrier (to give one example, the addition of transport inhibitors negatively impacted the integrity of the tissue). However, we were able to show the significant impact of adding an excess of untagged anti-LepR NF on the transport of the same tagged NFs, which reinforce the idea of a transport mediated by the binding to the LepR. Moreover, we confirm by ELISA the preserved binding activity of the NFs after transport, suggesting their preserved structural integrity. To finish, concurrently, we are developing reliable in vitro models designed for higher throughput screening of NFs and experimental conditions. With these, we would be able to draw more easily the mechanism of NFs transport. Any interesting outcomes will be shared in a subsequent publication.

Comment 4: Thank you for your insightful inquiry. Nanofitins possess the unique capability to non-covalently bind to a diverse range of well-expressed transporters and receptors within the intestine. In this study, our focus deliberately centered on the leptin receptor. The Nanofitins (NFs) featured in this paper were specifically designed through the ribosome display process to target the leptin receptor, as detailed in the Materials and Methods section. This receptor was chosen based on earlier in vitro proof-of-concept experiments conducted by P. Cammisoto et al., which highlighted the leptin receptor's proficiency in activating the specific receptor-mediated mechanism. Although we explored Nanofitins' potential with two additional receptors, our evaluations emphasized the leptin receptor's superior relevance and promise for our study objectives. Consequently, our intentional choice of the leptin receptor was predicated on its demonstrated efficacy in triggering the desired receptor-mediated mechanism. We aim to promptly share new findings with the scientific community, extending our investigation to a broader range of intestinal targets, as our ongoing efforts focus on enhancing the throughput of both ex vivo and in vitro assays.

Reviewer 4 Report

Comments and Suggestions for Authors

The authors describe a novel technology to transport proteins/peptides across biological membranes, also called Nanotitins. The article focuses on the characterization of the Nanofitins targeting the LepR (including binding characterization and cross reactivity in vitro) as well as ex vivo permeability studied across porcine intestinal tissue in an Ussing chamber setup. The article is very well-written and well-structured, which makes it easy to follow and understand the experimental work carried out and most of the discussion points the authors make. A few minor comments should be considered before publication:

Comments:

1)      In general, it is not completely clear which biological molecules/proteins are transported, and how these are “loaded into” or incorporated into the Nanofitins, which is the impression given throught the article (see examples below). This should be clearly stated in the introduction, the methods as well as the results.

Examples:

Line 19: “transport of robust affinity proteins, known as Nanofitins”

Line 89-91: “Ex vivo we successfully demonstrated the potential of an anti-LepR nanofitin candidate as an efficient carrier for transporting of functional cargo proteins….”

Line 685: “…to illustrate the transportation of other biological molecules (Figure 4 C).”

2)      The equations on page 9 should be named “Eq.1” and “Eq.2” with an appropriate caption.

3)      Are all the results illustrated in Fig2A-C carried out in a single replicate? This should be specified in the figure caption, as it was done for all the other illustrations.

4)      For the results section in general, it would be helpful for the reader if the graphs were shown before the detailed description of the results. For example for section 3.3.2, The figure could be introduced in line 573 (“…, were tested by ELISA (Figure 5).”, and the figure could then be shown shortly after, before starting the very detailed description of the results from line 577. It is very difficult to follow the results, without seeing the figure next to.

5)      In general, it should be checked that all expressions from latin are written in italics, e.g. “ex vivo”. This is not the case in several places through the manuscript.

6)      As a general discussion points, the authors should consider how Nanofitins would be formulated and delivered. The authors state that they would be useful for oral delivery, but how could it be imagined that they are encapsulated/protected? In a device-like carrier? Or a vehicle? How many Nanofitins would be needed to achieve a sufficient oral dose? How would it be realistic to achieve this amount in one device/formulation?

Author Response

Dear Reviewer,

Thank you for your valuable comments and suggestions. We appreciate the thorough review of our manuscript. Here are our responses and actions taken in light of your feedback:

Comment 1: We acknowledge the necessity for clarification regarding the transported biological molecules.

Each Nanofitin-derived construct mentioned in this study consists of a single polypeptide chain expressed in E. coli. This allows for the creation of either monomeric Nanofitins (i.e., single domain) or multimeric Nanofitins (i.e., two domains linked by a covalent polypeptide).

Except for the fluorescent labeling achieved through chemical conjugation of a dye, as detailed in the Materials and Methods section, there are no additional layers of loading, encapsulation, or incorporation. As a result, the final constructs comprise simple proteins. These include the moiety responsible for receptor-mediated transcytosis (1-F08 or 1-D07, anti-LepR domain), the moiety providing a biological activity such as half-life extension (NF2, anti-HSA domain), or irrelevant Nanofitins used as controls (irrNF, non-binding domain).

We have bolstered the manuscript by explicitly delineating the types of transported molecules in the introduction, methods, and results sections when applicable.

Comment 2: Thank you for pointing out this improvement. In alignment with the guidelines for authors of the journal Pharmaceutics, we have retained the original equation numbering as it appeared in the manuscript. However, for enhanced clarity, we have added annotations in the text referring to "equation 1" and "equation 2" corresponding to the equations on page 9. This adjustment aims to facilitate easier reference and understanding for the readers.

Comment 3: We appreciate your attention to detail. The results illustrated in Fig 2A-C were indeed carried out in a single replicate. We specified this information in the figure caption for transparency. The panel 2C was also improved to increase its readability.

Comment 4: Thank you for pointing this out. We agree with your suggestion. To enhance the reader's understanding, we have, accordingly, reorganize the presentation of graphs in the results section. Figures are now introduced before the detailed descriptions, providing a more accessible structure for the readers.

Comment 5: We thoroughly reviewed the manuscript and ensure that all Latin expressions are consistently written in italics throughout the document.

Comment 6: Thank you for your insightful input. Addressing your inquiries aligns precisely with our upcoming evaluation focus. As mentioned in the discussion, translating our study's findings to in vivo drug bioavailability and subsequent therapeutic effects requires further exploration. We anticipate publishing this year the outcomes of ongoing in vivo studies conducted with non-protected Nanofitins.

Regarding the rationale behind orally administrable Nanofitins used in first intention, several key points elucidate their potential. Firstly, previous demonstrations indicated that orally administered Nanofitins can exert a localized effect in the colon (pending publication, patent WO2016062874A1, European Project "SADEL" Scaffolds for Alternative DELivery, presented at the 11th ECCO Congress 2016). Secondly, Nanofitins, particularly those based on 1-F08 as presented in this study, can achieve concentrations exceeding 100 mg/ml, facilitating a liquid/gel formulation for oral administration at sufficient doses.

Finally, escalating doses of NFs in a buffered solution are currently administered via gavage to mice, ranging from 1 mg/kg up to 90 mg/kg. This approach aims to demonstrate the quantitative relevance of anti-LepR Nanofitins and determine if additional protection is necessary. When these data are gathered, we intend to rigorously evaluate our results, particularly to ascertain whether receptor-mediated transcytosis yields sufficiently potent effects to achieve therapeutic doses (validation of the chosen biological pathway). In the meantime, we are proactively engaging with specialists to anticipate potential requirements for safeguarding our molecules against the harsh gastrointestinal environment (validation of the formulation and administration route). Strategies under consideration include gastro-resistant encapsulation, probiotic-like approaches, or micro-needle devices, complementing the inherent high stability of Nanofitins against pH, chemicals, and temperature stimuli.